*A Nature Portfolio journal*

# Predicted action-effects shape action representation through pre-activation of alpha oscillations
Xin Wang[1], Shitao Chen[1], Keyang Wang[1] & Liyu Cao [1,2] ✉

Actions are typically accompanied by sensory feedback (or action-effects). Action-effects, in turn, influence the action. Theoretical accounts of action control assume a pre-activation of action-effects prior to action execution. Here we show that when participants were asked to report the time of their voluntary keypress using the position of a fast-rotating clock hand, a predictable action-effect (i.e. a 250 ms delayed sound after keypress) led to a shift of visuospatial attention towards the clock hand position of action-effect onset, thus demonstrating an influence of action-effects on action representation. Importantly, the attention shift occurred about 1 second before the action execution, which was further preceded and predicted by a lateralisation of alpha oscillations in the visual cortex. Our results indicate that when the spatial location is the key feature of action-effects, the neural implementation of the action-effect pre-activation is achieved through alpha lateralisation.

Actions are accompanied by sensory feedback. For example, knocking at a table produces a sound. According to the theory of event coding, actions and action-effects (i.e. the sensory feedback) are commonly coded in the cognitive system[1,2]. An action and its associated action-effects can form an event-file, which is activated during the action planning phase. The pre-activation of event-files has a large impact on both action control and the processing of sensory feedback, as demonstrated by a vast body of studies on this topic[3–8].

Action-effects have a strong impact on the representation of actions. This is demonstrated by the action-binding effect[9,10]. The action-binding effect refers to the observation that an action is reported to occur later when it is followed by a brief-delayed feedback (e.g. a sound follows an action by a delay of 250 ms; instrumental condition) than when no such feedback is given (baseline condition). In action-binding experiments, participants watched a clock with a clock hand that rotated rapidly in a clockwise direction. The action time was reported as the position of the clock hand when the action was made. The feedback (i.e. action-effect) in the instrumental condition is completely irrelevant to the task of action time reporting. Therefore, the feedback can be and should be ignored to achieve good performance in the task. Yet, the reality is that the influence of the feedback on action time reporting is unavoidable[9,11–13]. The action-effect seems to become an integral part of the action. Formulated under the framework of the theory of event coding, action and action-effect form an event-file.

Recent advances suggest that the action-binding effect is strongly influenced by attention[14,15], particularly a forward shift of visuospatial attention in the instrumental condition[16]. Two locations on the clock are particularly relevant. One is the clock hand location when the action was made (Location A; e.g. the 12 o'clock position). The other is the clock hand location when the sound feedback was played (Location S; e.g. the 2 o'clock position). Location S is at a future location relative to Location A since the sound feedback was played after the action. An objective observer should pay full attention to Location A and report Location A as the action time in both instrumental and baseline conditions. However, the sound feedback in the instrumental condition (Location S) also attracts attention and therefore leads to a forward shift of the distribution of visuospatial attention compared to the baseline condition[16]. Since the distribution of visuospatial attention is directly linked to the report of the clock hand position, a forward shift of visuospatial attention in the instrumental condition is translated into a forward shift in the reported clock hand position, which is the later reported action time in the action-binding effect. Yet, the time course of the visuospatial attention shift caused by action-effects remains unclear. Previous studies suggest that action binding is mainly driven by a prediction of action-effects[17,18], alluding to a shift of visuospatial attention prior to action onset.

Grounding action binding to a visuospatial attention basis not only helps to elucidate the cognitive mechanism behind the effect but also opens a possibility to study the neural processes related to the theory of event coding. This is because visuospatial attention has a robust neural indicator at the systems neuroscience level. Visuospatial attention to one hemifield of the visual field is associated with an increase of ipsilateral and a decrease of contralateral alpha power[19–22]. As introduced earlier, the critical feature of action-effects in the action-binding effect is the spatial location of the clock

[1]Department of Psychology and Behavioural Sciences, Zhejiang University, Hangzhou, China. [2]The State Key Lab of Brain-Machine Intelligence, Zhejiang University, Hangzhou, China. ✉e-mail: liyu.cao@zju.edu.cn

hand associated with action-effect onset. If the action-effect is pre-activated prior to the action execution, we should see a change in the attention paid to the spatial location associated with the action-effect. What is the neural signature and the time course of action-effect-related attention activation during a voluntary action?

In the present study, we first established behavioural evidence that action binding can be traced back to its origin as a shift of visuospatial attention when measured with the clock method. That is, when an action and its action-effect occupied different positions on a clock, the visuospatial attention paid to the action (Location A) was shifted to the location of its action-effect (Location S). Behavioural results further showed that the shift of visuospatial attention occurred at the onset of action, not after action execution, and only when a stable association was established between the action and action-effect (i.e. when an event-file was formed). Finally, electrophysiological data extended the behavioural results by showing that the shift of visuospatial attention started before the action onset and was preceded by an even earlier alpha lateralisation in the visual cortex. Therefore, the lateralisation of alpha oscillations implemented the action-effect-related attention shift prior to the action execution.

## Results

### Predicted action-effects lead to an attention shift at the onset of action

Participants watched a clock face with a fast-rotating clock hand (Fig. 1). After making a self-paced keypress, they were asked to either report the keypress time by giving the clock hand position (time reporting trials) or indicate if a visual probe was detected (attention trials). Visual probes were threshold-titrated disks. They were presented at the same time of keypress onset, at positions between −70° and 70° relative to the clock hand position at the keypress onset (Fig. 2a). In Experiment 1, when the keypress was followed by a 250 ms delayed sound feedback (instrumental condition), the distribution of visuospatial attention was more action-effect oriented than when the keypress was not followed by a sound feedback (baseline condition) (Fig. 2b). That is, attention was distributed more towards the future positions of the clock hand relative to the position at the time of the measurement, which was indicated by a significant interaction effect in a two-way (condition and probe location) within-participants ANOVA comparing the detection rate of visual probes ($F(7, 168) = 2.58$, $p = 0.033$, $\eta_p^2 = 0.10$). Note that the sound feedback in the instrumental condition was completely irrelevant to the task performance. Therefore, sensory feedback shifted action-related attention distribution to the direction of sensory feedback itself, as if the sensory feedback was part of the action. Consistent with this change in attention distribution, the reported action time was also more action-effect oriented in the instrumental condition ($M = 12.02$ ms, 95% CI = [−1.73 25.77] ms) than in the baseline condition ($M = -7.56$ ms, 95% CI = [−25.55 10.43] ms; $t(24) = 3.67$, $p < 0.001$, $dz = 0.73$; Fig. 2c). The forward shift of timing reports (i.e. the action-binding effect) and attention distribution in the instrumental condition was only observed when the instrumental and baseline conditions were tested in separate blocks (i.e. when the sensory feedback was predictable), but not with a mixed design (i.e.

when the sensory feedback was not predictable; Fig. 2d–f). When the two conditions were mixed in the same block in Experiment 2, no differences in the pattern of attention distribution ($F(7, 168) = 0.70$, $p = 0.618$, $\eta_p^2 = 0.03$) or timing reports were found ($t(24) = 0.13$, $p = 0.895$, $dz = 0.03$; instrumental condition: $M = -11.42$ ms, 95% CI = [−27.98 5.14] ms; baseline condition: $M = -10.81$ ms, 95% CI = [−22.53 0.92] ms). Thus, sensory feedback was integrated into action representation only when it was predictable.

### No attention shift observed after the action execution

We next asked if the attention shift observed at the time of action onset was also present after the action. Experiment 3 was identical to Experiment 1 except that the attention measure was taken at 150 and 350 ms after the keypress (Fig. 2g). Even though the sound feedback was still predictable in the instrumental condition as in Experiment 1, the attention distribution pattern was not statistically different between instrumental and baseline conditions at time points 150 ms ($F(7, 182) = 0.83$, $p = 0.538$, $\eta_p^2 = 0.03$) or 350 ms ($F(7, 168) = 1.02$, $p = 0.406$, $\eta_p^2 = 0.04$) after the keypress (Fig. 2h). Unsurprisingly, the reported action time was still more action-effect oriented in the instrumental condition ($M = 80.93$ ms, 95% CI = [59.69 102.17] ms) than in the baseline condition ($M = 59.07$ ms, 95% CI = [39.31 78.84] ms; $t(26) = 3.31$, $p = 0.001$, $dz = 0.64$; Fig. 2i). Therefore, there was still an action-binding effect in Experiment 3.

### Visuospatial attention and timing reports are tightly linked

The distribution of visuospatial attention at keypress onset and the reported keypress time both showed a forward shift pattern in the instrumental condition compared to the baseline condition. Using the attention data as input, we predicted the reported keypress time through cognitive modelling. The reported location of the clock hand at the keypress onset was modelled as the integration of each location weighted by the associated amount of attention (operationalised as the detection rate). The data modelling can reproduce the experimentally measured difference of timing reports between instrumental and baseline conditions. In Experiment 1, where the timing reports and attention distribution were both action-effect oriented in the instrumental condition, the modelled keypress timing reports were also more action-effect oriented in the instrumental condition ($M = -5.86$ ms, 95% CI = [−20.07 8.36] ms) than in the baseline condition ($M = -33.13$ ms, 95% CI = [−47.36 −18.89] ms; $t(24) = 3.60$, $p = 0.001$, $dz = 0.72$; Fig. 3a). In Experiment 2, where no significant difference was found in either the timing reports or the attention distribution, the modelled keypress timing reports were also not statistically different ($t(24) = 0.30$, $p = 0.764$, $dz = 0.06$); instrumental condition: $M = -21.69$ ms, 95% CI = [−36.62 −6.76] ms; baseline condition: $M = -23.96$ ms, 95% CI = [−40.51 −7.40] ms). Furthermore, the difference of the modelled timing reports between instrumental and baseline conditions positively correlated with the experimentally measured difference in timing reports (combining data from Experiments 1 and 2: $r(43) = 0.43$, $p = 0.003$; 5 outliers removed, Fig. 3b). However, with the attention data measured after the keypress, the data modelling results did not match the experimental findings. In Experiment 3, where an action-

**Fig. 1 | The task. a** In each trial, participants made a voluntary keypress. The keypress was either followed by a 250 ms delayed sound (instrumental condition) or not (baseline condition). After the keypress, they should report the keypress time or report if a visual probe was detected. **b** The time at the keypress onset was defined as time 0, and the corresponding clock hand position was defined as 0°. The diagonal line shows the clock hand position as it moves around the clock, with three example positions on the right (the keypress was made when the clock hand was at the 12 o'clock position in this case). The two examples of visual probe (the small white dot) were given on the top.

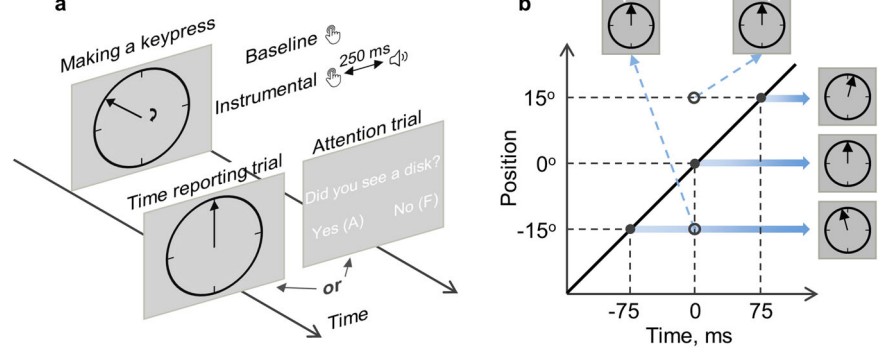

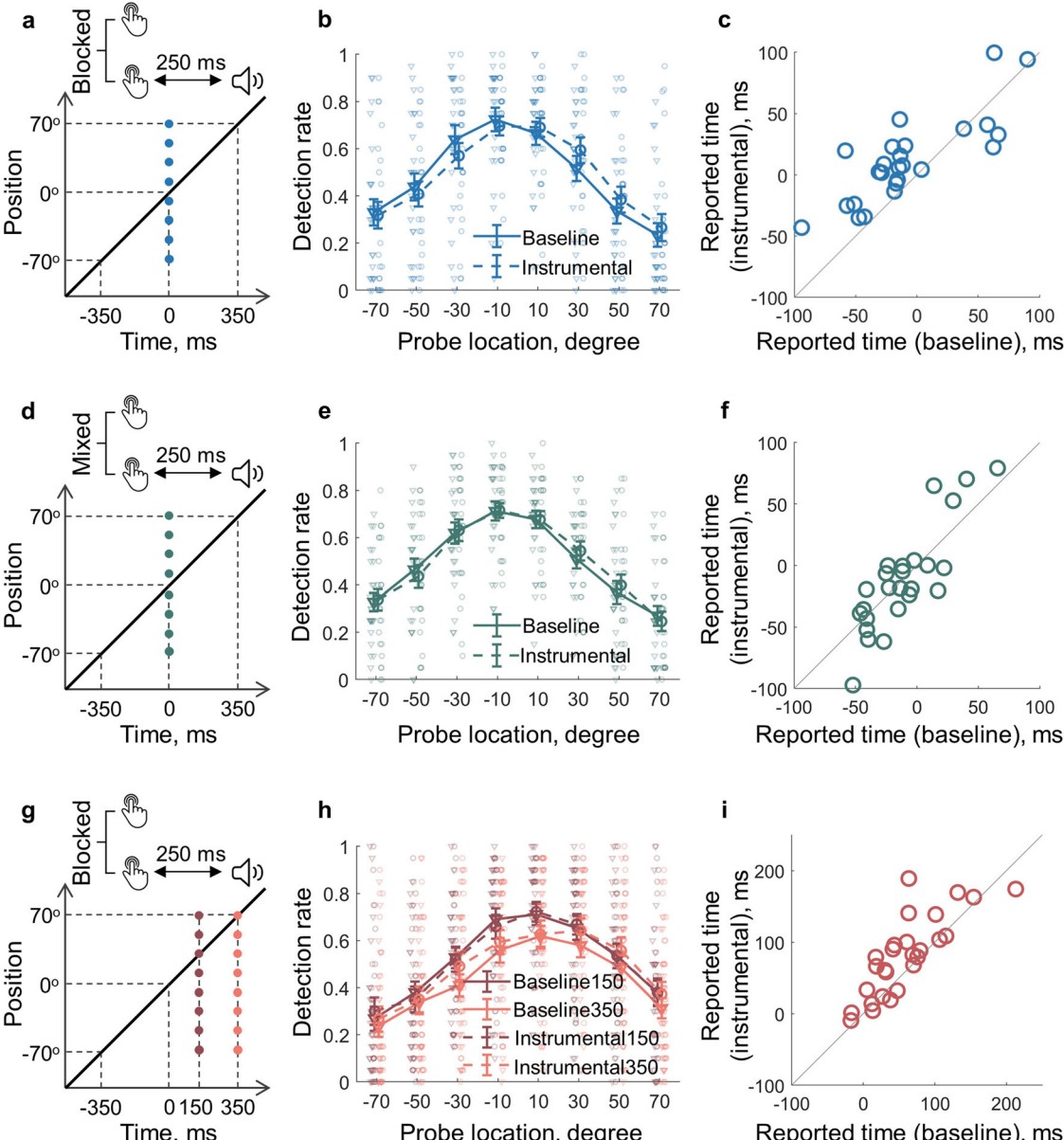

**Fig. 2 | Results from behavioural experiments. a–c** Results from Experiment 1 (*n* = 25 participants). **a** The visual probe was always presented at the keypress onset, but at one of eight spatial locations centring at the 0° location as shown in Fig. 1b. The instrumental and baseline conditions were tested in separate blocks. **b** The attention distribution was shifted towards the future in the instrumental condition compared to the baseline condition. **c** The reported keypress time was also later in the instrumental condition. Each circle represents a participant. **d–f** Results from

Experiment 2 (*n* = 25 participants). When the instrumental and baseline conditions mixed in the same block (i.e. the sound feedback was not predictable), neither the attention distribution nor the reported keypress time showed difference between the two conditions. **g–i** Results from Experiment 3 (*n* = 27 participants). When the sound feedback was predictable as in Experiment 1, no attention shift in the instrumental condition was found at 150 or 350 ms after the keypress compared to the baseline condition. Error bars represent ±1 standard error.

effect-oriented timing reports in the instrumental condition were found, the modelled keypress timing reports were not statistically different between instrumental and baseline conditions using attention data measured at 150 ms after the keypress ($t(26) = 0.78$, $p = 0.441$, $dz = 0.15$) or 350 ms after the keypress ($t(26) = 0.37$, $p = 0.714$, $dz = 0.07$) as input. This shows that attention after the keypress likely did not contribute much to the timing reports, which is also consistent with the finding of comparable patterns of attention distribution between instrumental and baseline conditions measured at both time points after the keypress.

### Lateralised alpha oscillation was associated with the attention shift before action execution

Behavioural results showed that predictable action-effects lead to an attention shift at action onset but not after action execution. We next sought

to investigate the attention distribution before action execution using EEG (electroencephalogram) in Experiment 4. Participants watched a clock face with a rotating clock hand and were asked to make a keypress when the clock hand was right at the 12 o'clock position (Fig. 4a). The keypress was either followed by a sound (instrumental condition) or not followed by a sound (baseline condition). Like in Experiment 1, the instrumental and baseline conditions were tested in separate blocks. To continuously track the distribution of visuospatial attention, we used flickering stimuli. A flickering stimulus of a specific frequency can trigger steady state visually evoked potentials (SSVEPs) at the same frequency. The amplitude of SSVEP can be used as an index of the amount of attention paid to the location of the flickering stimulus. Two flickering stimuli of different frequencies (13 and 15 Hz) were used, one on the top left and one on the top right of the clock face (Fig. 4a). The left frequency and the right frequency were

**Fig. 3 | Results of data modelling. a** With the attention data as input, the modelled action-binding effect matched the attention data. When a difference in attention distribution was found between the instrumental and baseline conditions (Exp 1), a significant action binding was obtained through modelling. When no such difference was found in attention distribution (Exp 2 and 3), the data modelling produced no significant action binding. The central mark of the boxplot is the median. The edges of the box are the 25th and 75th percentiles. The whiskers extend to the most extreme data points within 1.5 times the interquartile range. Data points outside 1.5 times the interquartile range are marked with crosses. **b** With the attention data measured at the keypress onset, the modelled action binding was positively correlated with the experimentally measured action binding.

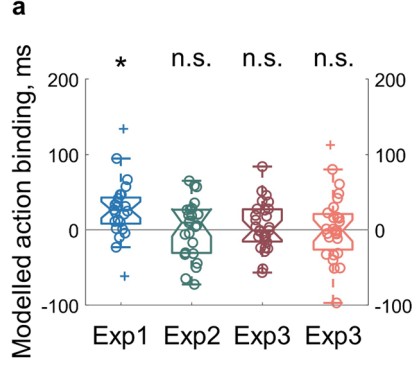

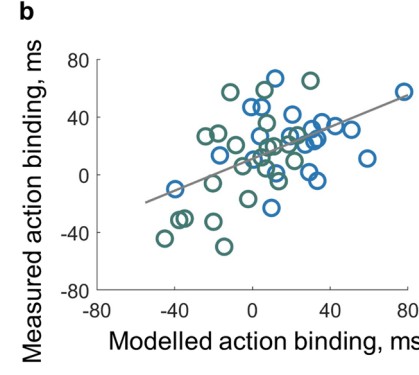

**Fig. 4 | Attention results from the EEG experiment. a** The task was to make a keypress when the clock hand was at the 12 o'clock position ($n = 27$ participants). Two large SSVEP stimuli of different frequencies (13 and 15 Hz; illustrated by the black and white colour) were presented during the task. **b** The action-binding effect of each individual. **c** SSVEP stimuli from left and right visual fields showed lateralised response patterns and were localised in the visual cortex. The SSVEP-derived attention index showed that the attention distribution was more towards the right (i.e. the future direction) in the instrumental condition as compared to the baseline condition, starting from about 1000 ms before the keypress and lasting until about 100 ms after the keypress (time points showing significant differences were marked with black lines). **d** The eye fixation data during the experiment. A significant difference was only found in the vertical axis between −1100 ms and −750 ms before the keypress onset (marked with a black line), with the average fixation position being 3.4 pixels lower on the screen in the baseline condition than in the instrumental condition.

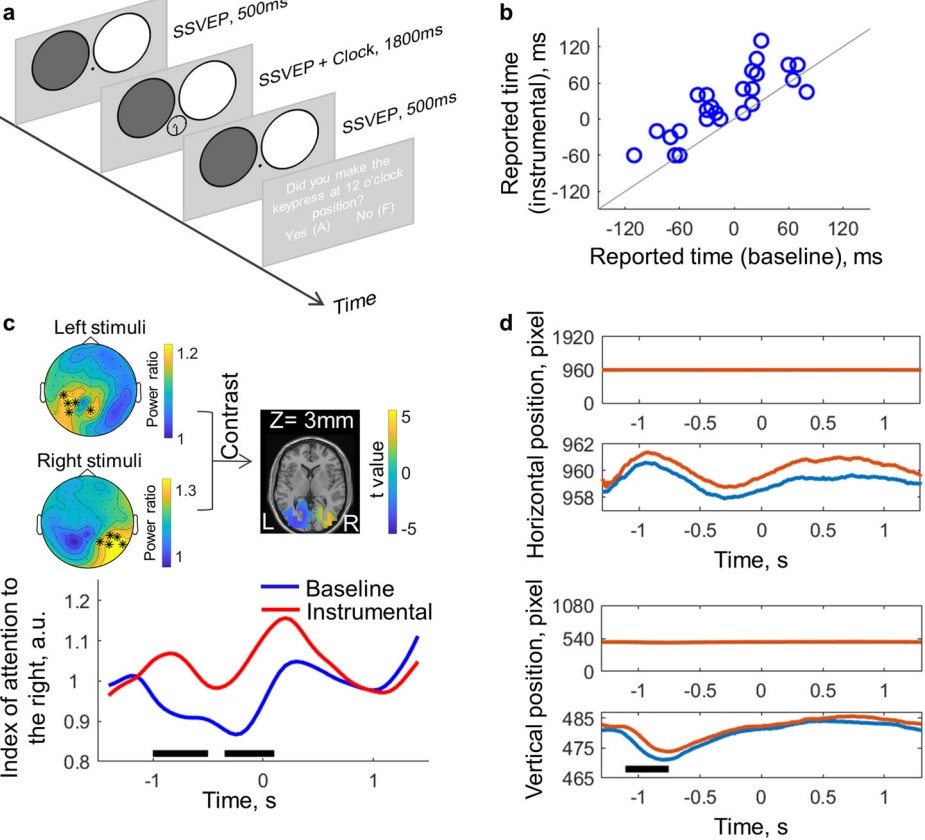

counterbalanced between participants. We also made sure that participants fixated at the 12 o'clock position on the clock during testing with eye-tracking.

The reported keypress time (i.e. the 12 o'clock position), relative to the actual keypress time, was more action-effect oriented in the instrumental condition ($M = 27.59$ ms, 95% CI = [9.00 46.18] ms) than in the baseline condition ($M = -7.96$ ms, 95% CI = [−26.13 10.21] ms; $t(26) = 6.29$, $p < 0.001$, $dz = 1.21$; Fig. 4b), replicating the action-binding effect. Flickering stimuli from the left visual field elicited SSVEP responses on the right hemisphere, and vice versa (Fig. 4c), demonstrating the effectiveness of the SSVEP stimuli. We also performed a source localisation analysis of the SSVEP responses. Comparing the source activation of left visual field stimuli and the right visual field SSVEP stimuli, we found significant positive

differences mostly in the right visual cortex and significant negative differences mostly in the left visual cortex (Fig. 4c; see Supplementary Fig. 1 for the full source localisation results), indicating that the SSVEP responses were mainly from the visual brain area. Using the five electrodes showing the strongest SSVEP response to the left SSVEP stimuli and five showing the strongest SSVEP response to the right (highlighted in Fig. 4c), we computed the index of attention to the right using the ratio between SSVEP responses to the right visual field and SSVEP responses to the left visual field (obtained through time-frequency analysis). Within each participant, the larger the index of attention to the right, the more attention is paid to the right visual field as compared to the left visual field. The index of attention to the right had a V shape before the keypress onset (time 0). It first went down, and gradually increased right before the keypress onset (Fig. 4c). This indicates

that before the keypress execution, participants first increased their attention to the left visual field and then shifted more attention to the right visual field before the keypress, which makes sense as the clock hand rotated from left to right around the keypress time. Importantly, when the keypress was followed by a predictable sound feedback (instrumental condition), more attention was paid to the right visual field than when the keypress was not followed by a sound feedback (baseline condition). This difference in attention distribution started from about 1000 ms before the keypress onset and lasted until 100 ms after the keypress, and was not caused by the loci of fixation (Fig. 4d). There was no significant difference in the horizontal axis of fixation position. Actually, the eye fixation position was numerically more towards the left in the instrumental condition as compared to the baseline condition, which was in the opposite direction to the attention difference. An index of attention difference was obtained for each participant by taking the average difference in the index of attention to the right between conditions over the time points showing a significant difference.

We then explored the neural underpinnings of the attention difference by comparing the neural oscillations between instrumental and baseline conditions in the pre-keypress time window. The power spectrum of the EEG data in the 3000 ms time window before the keypress was obtained. After averaging over trials and electrodes, a significant difference in the alpha band was found, with the instrumental condition having stronger alpha power than the baseline condition (Fig. 5a). With time-frequency analysis, the same comparison revealed stronger alpha power in the instrumental condition than in the baseline condition between −2500 and −900 ms before the keypress onset (Fig. 5b). However, the alpha power difference showed two peaks. The first peak was before −2000 ms. The topography of the alpha power difference between −2500 and −2000 ms showed a bilateral distribution in the parietal-occipital area, with the source localised mostly in the bilateral visual cortex (see Supplementary Fig. 2 for the full source localisation results). The second peak was around −1300 ms. The topography of the alpha power difference between −1800 and −900 ms showed right lateralisation, with the source localised mostly in the right visual cortex (see Supplementary Fig. 3 for the full source localisation

results). Note that the onset of the SSVEP stimuli and the clock was about −1400 and −900 ms, respectively. The alpha power difference peak at −1300 ms was right after the onset of the SSVEP stimuli. The temporal dynamics of alpha power showed that the alpha power increased in both conditions during the time of the second peak (Fig. 5c) and that the increase was stronger in the instrumental condition. A cross-participants correlation analysis was performed between the index of attention difference between instrumental and baseline conditions measured with SSVEP and the oscillatory power difference (averaged over the 5 electrodes showing the strongest alpha difference between −1800 and −900 ms; highlighted in Fig. 5b) in the pre-keypress time window. Strikingly, a significant correlation was only found in the alpha band overlapping the time period showing a stronger alpha power in the instrumental condition (Fig. 5d). Participants showing a larger alpha power difference between −1600 and −600 ms also showed stronger action-effect oriented attention shift between −1000 and 100 ms (Fig. 5e). Thus, the right-lateralised alpha oscillations in the visual cortex showing a peak around 1300 ms before the keypress onset likely implemented the attention shift.

## Discussion

In the present study, participants were asked to report the time of a self-initiated keypress through indicating the position of a fast-rotating clock hand at the time of the keypress. The reported keypress time was later when the keypress was predictably followed by a 250 ms delayed sound than when no-sound feedback was provided (i.e. the action-binding effect). We found that action binding was caused by an action-effect-oriented shift of visuospatial attention in the clock method used here. The action-effect-oriented shift of visuospatial attention was found from about 1000 ms before the action execution to the point of action onset and was only present when the action-effect was predictable. The attention shift was preceded and predicted by an even earlier alpha lateralisation, which peaked around 1300 ms before the action. Therefore, predictable action-effects caused an alpha lateralisation far before the action execution, which led to the shift of visuospatial attention and eventually the temporal action-binding effect.

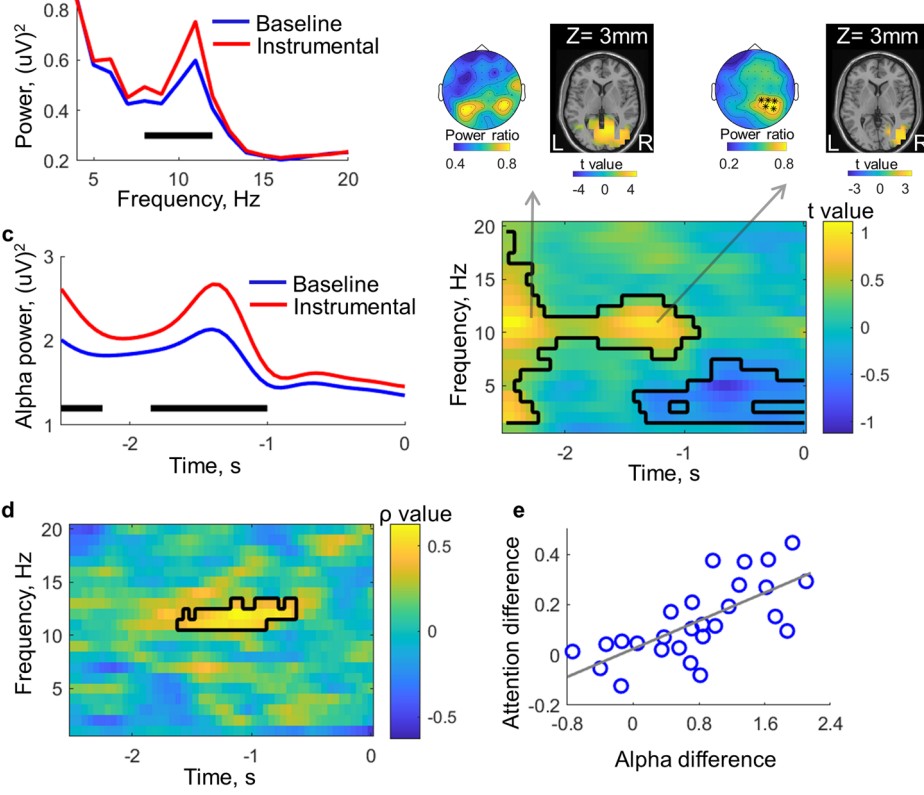

**Fig. 5 | Alpha power results from the EEG experiment. a** The instrumental condition had higher alpha power in the 3-s time window prior to the keypress than the baseline condition. **b** Time-frequency representation of the power difference between the two conditions in the pre-keypress time window. Positive values indicate stronger responses in the instrumental conditions. The alpha power difference between −1800 and −900 ms was right lateralised and localised in the visual cortex. **c** The temporal dynamics of alpha power. Note that the power scale was larger than the power scale in (**a**), which was due to the properties of the taper normalisation as implemented in the analysis software. **d** Correlation results between the attention difference derived from the SSVEP response and the power difference between the two conditions. **e** The scatter plot of the significant correlations as shown in (**d**), Spearman's $\rho = 0.76$, $p < 0.001$.

Predictable action-effects have a tremendous impact on action control[4,6,8,23]. A key idea behind such influence is that the representation of action-effects is activated prior to the action execution. For example, the forward model in action control assumes that the action-effect pre-activation is achieved through sending a copy of the motor command to relevant sensory brain areas[24–28]. In the framework of event coding theory, the action and associated action-effects form an event-file and are commonly coded in the cognitive system. The event-file is activated (or retrieved) prior to the action execution[1,5,7]. There is increasing experimental evidence in humans, from both functional magnetic resonance imaging[29] and neurophysiological[30,31] studies, supporting the idea of action-effect pre-activation. For example, when a keypress consistently triggered an SSVEP stimulus at 6 Hz, a reduction in the 6 Hz EEG signal was found before the keypress, which was assumed to be the consequence of action-effect prediction[30,32]. Our study corroborates previous findings by showing that a lateralisation of alpha power in the visual cortex is a neural signature of action-effect prediction in this particular task. The action-effect was a sound in the present study, but the key feature of the sound relevant to the task performance was the clock hand position at the sound onset (i.e. a visual feature). Therefore, a pre-activation of the action-effect was evident in a visual feature-related neural signal. Specifically, alpha oscillations in the visual cortex are widely known for the role of supporting the allocation of spatial attention[33]. A lateralisation of alpha power before the action onset indicates the pre-activation of the spatial information of the action-effect. This finding is consistent with the observation that action-effect pre-activation operates in the neural structure responsible for sensory processing of the action-effect[29]. Furthermore, our finding is also conceptually consistent with a previous study showing that predictable visual action-effects were associated with increased rates of spontaneous saccades towards the location of action-effects before the action-effects were presented[34]. Importantly, we could exclude the possibility that the alpha lateralisation observed here was due to saccades through careful eye-tracking control.

Neural oscillations play a key role in regulating brain functioning[35–37]. A number of studies have provided converging evidence for the involvement of neural oscillations at different frequency bands in the processes of perception-action integration and action selection, thus giving neural oscillations an important role as the neural underpinnings of the theory of event coding[32,38,39]. For example, theta oscillations were found to be particularly relevant when there was a demand for the reconfiguration of event-files[32,38,40–42]. This is well in line with the low-frequency nature of theta oscillations, which offers a long time window for information integration over a broad range of brain areas. Alpha oscillations work in this process through information gating, that is, filtering out irrelevant features to the current event-file[36,38,39,43]. In this sense, the present study provides complementary evidence to the previous studies in a different testing task. Specifically, the increase of alpha power in the right visual cortex prior to action execution inhibited the attention to the left visual field, which was the attention side contrary to the action-effect (i.e. the information that should be avoided). The alpha power increase peaked around 1300 ms before the action execution, which was quite early. Given the complexity and flexibility of human actions, a mechanism that supports faster action initiation may be more preferable. However, the action required in the current study was a prepared ballistic keypress, making the early alpha change understandable. In this sense, our results echo the unconscious preparatory process of action control[44]. However, it remains unclear if the timing of action-effect pre-activation is flexible in more naturalistic everyday actions.

For the action-effect-oriented attention shift, both behavioural and EEG data showed that the attention shift was predictive. That is, the attention shift occurred before the action onset. This is consistent with previous findings that the action-binding effect was largely caused by action-effect prediction in healthy individuals[17]. This does not necessarily mean that the action-binding effect receives no contribution from the so-called retrospective component (i.e. the sound feedback influences action time reporting even when it is not predictable). In the study by Moore and Haggard[18], the reported time of action with unpredictable sound feedback (as in the current Exp 2) was later when compared to a condition in which

action was never paired with a sound feedback. In the current study, we did not include such a no-sound condition to investigate the retrospective component of action binding. It should also be noted that the earliest time point (about 1000 ms before action onset) when attention shift started to emerge in the SSVEP measure should be taken with caution. The presentation of SSVEP stimuli started about 1400 ms before the action onset. Therefore, the time point when we could first obtain a reliable SSVEP response was about 900 ms before the action onset as a 1000 ms time window was taken for the power calculation. This means that the attention shift may have started even earlier.

To conclude, action-effect prediction led to a shift of visuospatial attention towards the action-effect. The attention shift occurred before the action execution and underlay the temporal action-binding effect measured with the clock method. Even earlier lateralisation of alpha oscillations before the action execution predicted the attention shift, which served as the neural process supporting the action-effect prediction.

## Methods
### Experiment 1
**Participants.** 30 participants (19 females; mean age = 21.7, SD = 2.8) were recruited from a local participant pool. With 30 participants, the smallest effect size that can be detected with a statistical power of 0.9 at the standard 0.05 alpha error probability (one-tailed) using a paired-sample $t$-test is 0.55 (calculated with GPower version 3.1.9.7[45]). The action-binding effect is estimated to have an effect size of about 0.70[16]. For the critical effect of an attention distribution difference between different conditions, the smallest effect size (partial eta squared) of an interaction effect with within-participants 2 by 8 ANOVA that can be detected with a statistical power of 0.9 at the standard 0.05 alpha error probability is 0.09 (calculated with MorePower version 6.0.1[46]). All participants had normal or corrected-to-normal vision. Written informed consent was obtained prior to the experiment, and participants were debriefed and received monetary payment after the experiment. The experiment was conducted in accordance with the Declaration of Helsinki (2013) and was approved by the Ethics Committee of the Department of Psychology and Behavioural Sciences, Zhejiang University (ethics application number: [2022] 003). All ethical regulations relevant to human research participants were followed.

**Stimuli, task and procedure.** The experiment was a within-participants design with two conditions included. In the instrumental condition, participants watched a clock on the screen. The clock had a clock hand that rotated rapidly in a clockwise direction. Participants made a self-paced keypress, which triggered a sound feedback with a delay of 250 ms. After the keypress, participants reported either the position of the clock hand when the keypress was made (time reporting trials) or if a visual probe was detected (attention trials), depending on the instruction on the screen. Therefore, participants did not know what to report in advance. In the baseline condition, everything was the same as in the instrumental condition except that no-sound feedback followed the keypress. The instrumental condition always followed the baseline condition. The testing order has been found to influence the size of action binding, and testing the baseline condition first produced a larger action-binding effect[47]. The testing order was not very relevant to the current study, as the main interest here was about the pattern of attention distribution (i.e. the relative amount of attention paid to each tested position reflected in visual probe detection rates) in each condition. Effects such as fatigue may influence the overall detection rate of the visual probe in a condition, but should not influence the relative detection rates at different positions in the same condition.

In each condition, there were 30 time reporting trials and 180 attention trials. For attention trials, 160 trials had a threshold-titrated visual probe that was always presented at the time of keypress onset, but at one of eight positions just outside the clock rim (20 trials for each position). The remaining 20 trials had no visual probe presentation (for evaluating false

alarms). The clock hand position at the keypress onset was defined as the 0° position. The eight probe presentation positions were: −70°, −50°, −30°, −10°, 10°, 30°, 50° and 70°. Negative position values indicate positions the clock hand pointed to before the keypress was made, and positive position values indicate positions the clock hand pointed to after the keypress was made. All trials were presented in a random order within each condition. Five practice trials were given for each condition.

In each trial, participants watched the centre of the clock and the clock hand started rotating from a random position. They were instructed to make a self-paced keypress with the right index finger (pressing 'k' on a standard QWERTY keyboard). The choice of keypress timing should be made without using any strategies except that the keypress should be made at least 1 s after the clock hand starts rotating. A trial would be aborted and repeated if a keypress was made too early. After the keypress, a 250 ms delayed sound feedback would be given depending on the testing condition, and the clock hand continued rotating for a random duration between 1 and 1.5 s. In time reporting trials, the clock hand would stop at the 12 o'clock position. Participants reported the keypress time by moving the clock hand to the position when the keypress was made using the left hand (pressing 'a' and 's' to move the clock hand counter-clockwise by 10° and 1°, respectively; pressing 'd' and 'f' to move the clock hand clockwise by 1° and 10°, respectively). In attention trials, a prompt appeared on the screen asking participants to indicate if a visual probe was detected. Participants made choices by pressing 'a' (detected) or 'f' (not detected). After the response was made, the next trial started with an inter-trial interval of 1 s. Since there were more attention trials than time reporting trials and participants did not know the exact task until the end of a trial, we instructed participants at the beginning of the experiment that they should perform the task as if the time reporting were required in each trial. This is to ensure that the time reporting results were of good quality for the data analysis.

Visual stimuli were presented on a liquid crystal display screen with a grey background (refresh rate: 100 Hz; 24-inch screen size). The clock face had a diameter of 2.7 degrees of visual angle (120 pixels). The clock hand rotated with a speed of 1800 ms per revolution. The visual probe was a disk presented for 30 ms with a diameter of 0.1 degrees of visual angle and a distance of 1.5 degrees of visual angle from the clock centre. The luminance threshold of the visual probe was obtained using a 2-down-1-up staircase procedure[48] in a threshold testing session prior to the main experiment. The sound feedback used in the instrumental condition was a 1000 Hz tone (50 ms long, 5 ms rise/fall envelope, comfortable volume level) presented with a pair of headphones (Beyerdynamic DT 770 pro, 32 OHM, Germany). Stimulus generation and presentation were controlled by Psychtoolbox-3[49] using Matlab (The MathWorks Inc., USA). The experiment was performed in a well-lit, soundproof testing booth.

## Experiment 2
A new group of 30 participants (15 females; mean age = 22.3, SD = 2.6) were recruited from a local participant pool. Experiment 2 was identical to Experiment 1 except that the instrumental and baseline conditions were mixed in the same block. That is, all the 420 trials (210 trials in each condition) were mixed and presented in a random order.

## Experiment 3
A new group of 30 participants (13 females; mean age = 23.0, SD = 3.2) were recruited from a local participant pool. Experiment 3 was identical to Experiment 1 except for the following three changes. First, attention measure was not taken at the keypress onset, but at 150 and 350 ms after the keypress onset. For each time point, the same eight positions were used (from −70° to 70° relative to the clock hand position at the keypress onset). Therefore, each condition contained 340 attention trials (2 time points × 8 positions × 20 trials each, plus 20 catch trials). Second, the number of time-reporting trials increased to 60 in each condition. Thus, each condition had 400 trials. Third, the ABBA design was used. Each condition was evenly distributed in two testing blocks. Each testing block contained 200 trials (2 time points × 8 positions × 10 trials each with visual probe presentation,

plus 10 catch trials, plus 30 time reporting trials). This led to two baseline blocks and two instrumental blocks for each participant. The four blocks followed the order of ABBA, with half participants starting with the baseline condition and the other half starting with the instrumental condition.

## Experiment 4
A new group of 30 participants (7 females; mean age = 23.5, SD = 3.9) were recruited from a local participant pool. The time reporting task used in previous experiments was modified for continuously measuring the distribution of visual spatial attention. Participants still watched the same clock with a fast-rotating clock hand. The task was to make a keypress when the clock hand was exactly at the 12 o'clock position[47]. If participants reported that a keypress was successful, it also indicated that the reported keypress time was when the clock hand at the 12 o'clock position. With this task design, the attention distribution at around the 12 o'clock position was particularly relevant. To continuously track the attention distribution, we included two flickering stimuli, one to the top left of the 12 o'clock position and the other to the top right. The flickering stimuli would elicit a frequency following response called the SSVEP, which could be measured with EEG. The SSVEP response has the same frequency as the flickering stimulus. The amplitude of the SSVEP response was used as an index of the amount of attention allocated to the location of the corresponding flickering stimulus. The more attention allocated, the stronger the SSVEP response.

The experiment was a within-participants design with a baseline condition and an instrumental condition. The only difference between the two conditions was that a keypress triggered a 250 ms delayed sound in the instrumental condition but not in the baseline condition. Each condition had 80 valid trials. Like in Experiment 1, the baseline condition was always tested first.

Participants pressed a space bar to start a trial. After the space bar keypress, the two flickering stimuli were presented immediately and lasted for 2800 ms. The clock appeared 500 ms after the space bar keypress, with the clock hand starting to rotate clockwise from the 6 o'clock position. When the clock hand rotated back to the 6 o'clock position (one lapse, 1800 ms), the clock and clock hand disappeared. Therefore, the flickering stimuli were presented 500 ms before the clock's appearance and lasted for another 500 ms after the clock's disappearance (Fig. 4a). Participants made a keypress (pressing 'k' on a standard QWERTY keyboard) with their right index finger when the clock hand was right at the 12 o'clock position. After the disappearance of the flickering stimuli, participants indicated if the keypress was made when the clock hand was at the 12 o'clock position (pressing 'a' with left hand) or not (pressing 'f' with left hand). A trial was counted as valid if the participant indicated that the clock hand was at the 12 o'clock position when the keypress was made. The average inter-trial interval was 6.64 s (SD = 1.22, range = [4.92–9.45]) in the baseline condition and 5.96 s (SD = 1.09, range = [4.71–10.04]) in the instrumental condition.

We also used eye-tracking (Eyelink Portable Duo, SR Research Ltd, Canada) to ensure that during the whole period of the flickering stimuli presentation, participants always fixated at the 12 o'clock position. If the eye fixation deviated from the required fixation point by 1 degree of visual angle (45 pixels on the screen), the trial would be aborted immediately. In the time period when the clock was not presented, a small white dot was displayed at the 12 o'clock position to facilitate compliance to the fixation requirements.

The parameters of visual stimuli were the same as in Experiment 1. The flickering stimuli were disks with 8.1 degree of visual angle in diameter. One flickering stimulus was 13 Hz, and the other was 15 Hz. Half participants had the 13 Hz flickering stimulus on the left, and the other half had the 15 Hz flickering stimulus on the left. The luminance of the flickering stimuli underwent rhythmic changes between full black ([0 0 0] in RGB space) and full white ([255 255 255]) in according frequencies.

EEG data were recorded with a BioSemi ActiveTwo system with 64-channel Ag/AgCl active electrodes placed in accordance with the 10–20 system. We also had two extra electrodes placed on the left and right earlobes for possible re-referencing and four extra electrodes for electro-oculogram monitoring (one below the left eye, one above the left eye and two

to the outer canthi). During the data recording, the online reference was composed of sites CMS and DRL, and the sampling rate was 1024 Hz. Electrode impedance was kept below 20 kΩ.

**Statistics and reproducibility**

For Experiments 1–3, the data analysis approach was similar. For time reporting trials, a reported time was calculated. For each trial, the difference between the reported position of the clock hand and the actual position of the clock hand at the keypress onset (a spatial difference) was converted to a reported time (i.e. time judgement error), based on the clock hand rotation speed of 1800 ms per revolution. For attention trials, the detection rate at each location was calculated as the ratio of trials with visual probes being detected among all the trials with a visual probe presentation at that location. The false alarm rate was calculated as the ratio of reporting detecting a visual probe in catch trials.

Participants were excluded if extreme values were found in the time reporting or attention tasks. For the time reporting task, the standard deviation and the median of timing reports in each condition were considered. Extreme values may indicate poor quality in the timing report data. For the attention task, the detection rate and the relative difference between the detection rate and false alarm rate (the difference between the two divided by the sum of the two) were considered. Extreme values may indicate very low detection rates or very high false alarm rates. Extreme values were detected using the rule of median absolute deviation from median (MAD−median): Let $p$ be the individual value and $P$ be the individual values from the whole sample. An individual value is an outlier if $|p - \text{median}(P)| \times 0.6745 > 3 \times \text{MAD} - \text{median}$[50]. In total, five participants from Experiment 1, five participants from Experiment 2, and three participants from Experiment 3 were excluded from further analysis.

The action-binding effect was evaluated with a paired-sample $t$-test comparing the median of the reported keypress time between the instrumental condition and the baseline condition (one-tailed in Experiments 1, 3 and 4, assuming a replication of action binding; two-tailed in Experiment 2, with no prediction on the direction of the effect). The pattern of attention distribution was evaluated by comparing the detection rate with a two-way (condition and probe location) within-participants ANOVA[51]. An interaction effect between condition and probe location would indicate different patterns of attention distribution between instrumental and baseline conditions.

In the modelling of timing reports with the detection rate data, the timing represented by each sampling location was weighted by its corresponding detection rate and then summed[16]. This was done as the following:

$$D'_k = D_k - \min(D) \tag{1}$$

$$\text{Modelled keypress timing report} = \sum_{(k=-350)}^{350} T_k \times \left(D'_k / \sum_{(k=-350)}^{350} D'_k\right) \tag{2}$$

where $k$ is the 8 attention sampling locations, $D_k$ is the detection rate at location $k$, $D$ is an array of detection rates from all the 8 locations, $\min(D)$ is the smallest among the 8 detection rates and $T_k$ is the reported timing if attention was fully focused on location $k$ (e.g. locations −70° and 70° correspond to −350 ms and 350 ms, respectively).

The modelled keypress timing reports were also compared between instrumental and baseline conditions with paired-sample $t$-tests (one-tailed in Experiment 1 as a significant difference in the attention distribution pattern was found between instrumental and baseline conditions; two-tailed in Experiments 2 and 3 as no significant difference in the attention distribution pattern was found).

We used Spearman's rank correlation analysis throughout the manuscript whenever a correlation analysis was performed. Bivariate outliers in correlation analyses were detected and removed using the box-plot rule[52].

For Experiment 4, the behavioural data were analysed similarly as in Experiment 1. A reported time was calculated for each trial. In this case, the reported position of the clock hand at the keypress onset was the 12 o'clock position for each trial. In rare cases, trials were excluded if no keypress was found (trial exclusion step 1).

The raw EEG data were band-pass filtered between 0.1 and 30 Hz using a windowed sinc FIR filter (one-pass with zero phase distortion, Kaiser window) and re-referenced to the average of the two earlobe electrodes before being segmented into trials. Three participants used the average T7 and T8 electrodes for re-referencing as the earlobe electrodes were extremely noisy. Each trial was 4500 s long, aligned to the keypress onset that was required to be made when the clock hand was at the 12 o'clock position (3000 ms before the keypress onset and 1500 ms after). Noisy electrodes were visually identified and replaced with the average of neighbour electrodes. Trials were excluded if the amplitude range between −1500 ms and 1000 ms exceeded 200 μV (trial exclusion step 2). After the two steps of trial exclusion, 3 participants were excluded from further analysis as the remaining trials (88, 76 and 13) were less than 100 out of the total 160 trials. With the 27 participants included in the final analysis, the average number of remaining trials was 147.1 (SD = 14.7).

We first calculated the power spectrum of the pre-keypress data ([−3000 0] ms) for each trial using a fast Fourier transform. The power spectrum (averaged across trials and the 64 on-scalp electrodes) was compared between the instrumental and baseline conditions using paired-sample $t$-tests between 1 and 20 Hz in steps of 1 Hz. We then performed time-frequency analysis on the single trial EEG data using a 1000 ms long sliding window in steps of 50 ms from −2500 ms to 900 ms. For each participant, a matrix of $t$ values was obtained by comparing the time-frequency data ([−2500 0] ms, [1 20] Hz) between the instrumental and baseline conditions using unpaired-sample $t$-tests (averaged across electrodes). Therefore, the $t$ value matrix showed the difference in the time-frequency data between the two conditions for each participant. The $t$ value matrices were compared to 0 for evaluating the statistical difference in the time-frequency data between the two conditions. The five electrodes (alpha electrodes: 'PO8', 'PO4', 'P6', 'P4' and 'P2') showing the largest difference in alpha power between −1800 ms and −900 ms were selected for the correlation analysis with the index of attention difference. Multiple comparisons in the statistical analyses above were corrected using the cluster-based permutation approach[53]. In short, clusters based on adjacency in the frequency or the time-frequency map were first formed with points showing significant differences between conditions (or significant correlations in the analysis below, with $p < 0.05$ before the multiple comparison correction). For each identified cluster, a cluster-level statistic was calculated as the sum of the test statistics at each point (e.g. the $t$ value in the case of a $t$-test). The original data were then randomly permuted between conditions assuming no difference (or between participants assuming no correlation in the analysis below) for 1000 times. For each permutation, the largest cluster-level statistic was kept. The original cluster-level statistics with values bigger than the 97.5 percentile or smaller than the 2.5 percentile of the cluster-level statistics obtained through permutation were considered statistically significant (i.e. a two-sided test).

For the SSVEP analysis, the raw EEG data were segmented into 4000 ms long trials without re-referencing or filtering. Trials excluded in the previous analysis were also excluded here. Time-frequency analysis was performed using a 1000 ms long sliding window in steps of 50 ms from −1400 ms to 1400 ms. For each trial and each electrode, a 13 Hz SSVEP signal was obtained at each time point between −1400 ms and 1400 ms in steps of 50 ms with reference to the average of 12 Hz and 14 Hz responses in the corresponding condition through division. Similarly, a 15 Hz SSVEP signal was obtained with reference to the average of 14 Hz and 16 Hz responses. We then obtained the topography of the SSVEP signal separately for the SSVEP stimulus on the left visual field and the SSVEP stimulus on the right visual field. Based on the topography, we selected five electrodes showing the strongest response to the left SSVEP stimulus ('P8', 'PO8', 'P10', 'P6' and 'PO4') and five electrodes showing the strongest response to the

right SSVEP stimulus ('POz', 'P5', 'P3', 'PO7' and 'CP5') for subsequent analyses. For each condition, an index of attention to the right was obtained by referencing the SSVEP signal induced by the stimulus from the right visual field to the SSVEP signal induced by the stimulus from the left visual field (through division). The index of attention to the right was averaged across trials and the selected electrodes before being compared between the instrumental and baseline conditions with paired-sample $t$-tests. An index of attention difference between conditions was obtained by taking the numerical difference of the index of attention to the right averaged over the time points showing significant differences between conditions. Correlation analyses were performed between the index of attention difference and the power difference between conditions (a matrix of $t$ values obtained by comparing the time-frequency power averaged over the alpha electrodes between conditions). Multiple comparisons were corrected using the cluster-based permutation approach[53].

Source localisation analysis was performed using the method of dynamic imaging of coherent sources (DICS)[54]. A template grid was computed based on a template head model (the standard Boundary Element volume conduction model) with a spatial resolution of 1 cm[55]. The AAL brain atlas[56] was then mapped onto the template grid, and the cerebellum was excluded in the subsequent source localisation analysis. For the SSVEP source localisation, SSVEP responses between [−1200 1200] ms and baseline activity in the same frequency between [−2500 −1500] from all trials were localised with a common filter using DICS (hanning taper; regularisation set at 15%). The SSVEP source activity was then referenced to the baseline source activity ((SSVEP−baseline)/(SSVEP + baseline)) separately for SSVEP stimuli from the left and right visual field. Finally, the source activity for left visual field stimuli was compared to the source activity for right visual field stimuli using paired-sample $t$-tests (cluster-based permutation correction). Therefore, it was expected that right-lateralised positive clusters and left-lateralised negative clusters should be found in the source space. For the alpha source localisation, alpha activities (10 Hz, ±3 Hz with discrete prolate spheroidal sequence filter) in the time window showing no lateralisation in the between-condition power difference ([−3000 −2000] ms) and in the time window showing a right lateralisation ([−1800 −800] ms) in the power difference were localised separately. For each time window, the alpha activities in the instrumental and baseline conditions were localised with a common filter using DICS (regularisation set at 15%). The alpha source activity was then compared between the instrumental condition and the baseline condition using paired-sample $t$-tests (cluster-based permutation correction). Significant clusters were projected to a template T1-weighted magnetic resonance image for visualisation (the 'colin27' brain[57]).

Horizontal and vertical components of eye-movement data were extracted and aligned to the keypress onset similarly as done for EEG data. Each trial of eye-movement data was between −1200 ms and 1200 ms (with the keypress at time 0). The original sampling rate of eye-movement data was 1000 Hz for 25 participants, 500 Hz for two participants and 250 Hz for three participants. The data were downsampled to 250 Hz before being compared between the instrumental and baseline conditions using paired-sample $t$-tests, separately for the horizontal and vertical components. Multiple comparisons were corrected using the cluster-based permutation approach[53].

The data analysis was performed with Matlab (The MathWorks Inc, USA) using the FieldTrip toolbox[58].

### Reporting summary
Further information on research design is available in the Nature Portfolio Reporting Summary linked to this article.

## Data availability
The raw data reported in this manuscript have been uploaded to Open Science Framework for free access (https://doi.org/10.17605/OSF.IO/QVX8W)[59].

## Code availability
The data analysis script has been uploaded to Open Science Framework for free access (https://doi.org/10.17605/OSF.IO/QVX8W)[59].

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

## Acknowledgements

We would like to thank Zongze Chen for assistance with data collection and Nathan Thomas Han for English editing. This work was supported by the National Natural Science Foundation of China (grant number: 32271078), Fundamental Research Funds for the Central Universities (grant number: 226-2024-00207) and STI 2030—Major Projects (grant number: 2021ZD0200409).

## Author contributions

X.W. performed the experiments and data analysis, revised the manuscript; S.C. performed the experiments and revised the manuscript; K.W. performed the experiments and revised the manuscript; L.C. conceptualised the study, designed the experiments, performed the data analysis, provided supervision and wrote the manuscript.

## Competing interests

The authors declare no competing interests.
