## [Transparent Peer Review file · Communications Biology]

Predicted action-effects shape action representation through pre-activation of alpha oscillations

Corresponding Author: Dr Liyu Cao

Version 0:

Reviewer comments:

Reviewer #1

(Remarks to the Author)

This paper seeks to find evidence for anticipatory codes of action effects, hence predictable perceptual feedback of motor activities (in this case keypresses). When participants were asked to report the timepoint of an action and its predictable auditory effect on a Libet clock, they perceived the action to occur later than when the same action produced no effect (temporal action binding). Importantly, this action binding went along with a shift of visual attention towards the position of the pointer at which an auditory effect predictably occurred. This attention shift was apparent already 1 s before action onset in detection performance for small visual stimuli around the Libet clock and it was apparent even earlier in EEG data.

This is an interesting manuscript. In fact, it is tricky but would be theoretically important to find precursors of perceptual feedback codes in action production. The paper does so in a clever way by combining and building on various sources of evidence. All methods appear very sophisticated to me. I have a few comments the authors may consider.

1) There have been other attempts to track the 'unobservable' (codes of forthcoming perceptual feedback). One approach relied on frequency tagging such effects, and checking whether there are power changes in the EEG corresponding to forthcoming action effects prior to action execution (Dignath et al., 2020 JEP:G). Another approach relied on anticipatory eye movement towards the location of upcoming action effects (Pfeuffer et al., 2016, JEP:G). The present approach is markedly different and novel, but the pros and cons might be discussed somewhere.

2) It was not perfectly clear to me why there was no attention shift apparent when attention was probed only 150 ms after the keypress, which is a very short delay, although there was still action binding present in that condition (Exp. 3). Also, the EEG study suggest an attention shift rightwards up to 700 ms after keypress onset. So would we not predict that the attention shift was apparent even 150 ms after the keypress in Exp. 3 as well? I think this apparent contradiction (at least in my eyes) should be discussed somewhere.

3) I missed one piece of information, but it may have escaped my attention. How long did the pointer rotate before participants made a self-paced keypress? I'm asking because the EEG data suggest a fairly long pre-activation interval. This would make sense only if the trial lasted already that long and did not include events from preceding trials.

4) Wording. In several places the authors talk about 'pre-activation of action-effects' (e.g. in the discussion, L. 283). I think this is not a 100% correct. What the data very convincingly show is that there is a shift of visual attention towards the position where a rotating pointer would arrive when a predicted auditory effect is presented. This indicates that the time point of the effect is somehow pre-activated. But what else of this auditory event is represented or pre-activated, we don't know. This would require to observe traces of features other than the time point of this auditory event (like pitch or intensity).

Minor observations

P1, l1, Should it read 'predicted action effects shape...'

L 31. Event-fileS

l. 60 A previous suggested ... the sentence reads odd, something is missing.

L 151 detection rate

L 231 field instead of filed.

L 284, Has shown instead of showed

Reviewer #2

(Remarks to the Author)

This is a well-written manuscript about a well-motivated study. The presentation is dense and not always easy to follow, but I think the logic and the sequence of the experiments makes sense, and is informative. The studies were carried out expertly, the findings are clear and interesting, and the discussion is reasonably balanced. My only concern relates to the fact that the authors make false claims regarding precedence: the fact that representations of action effects are activated before the action is initiated has been reported before, both for fMRI and EEG methods, see:

1. Kühn, S., Keizer, A., Rombouts, S.A.R.B., & Hommel, B. (2011). The functional and neural mechanism of action preparation: Roles of EBA and FFA in voluntary action control. *Journal of Cognitive Neuroscience*, 23, 214-220.

2. Dignath, D., Kiesel, A., Frings, C., & Pastötter, B. (2020). Electrophysiological evidence for action-effect prediction. *Journal of Experimental Psychology: General*, 149, 1148-1155.

This should be rectified: claims of precedence need to be downplayed and the remaining novel contribution of the present findings need to be discussed more specifically.

Signed, Bernhard Hommel

Reviewer #3

(Remarks to the Author)

The manuscript 'Predicted action-effect shapes action representation through pre-activation of alpha oscillations' by Wang and colleagues deals action-effects in an interesting and innovative way. Overall, I really enjoyed reading it, although I have a few remarks.

I feel like some parts of the manuscript are quite challenging and hard to understand. This might be due to the extent of reported experiments and the structure required by the journal (methods at the end of the manuscript). Although the authors seem to acknowledge this and have included quite a bit of methodology in the results section, the manuscript is hard to understand and the line of argument hard to follow. A thorough revision of the manuscript in terms of readability would make the manuscript more accessible for a wider readership.

In terms of the overall structure, it seems quite odd to me that the Introduction includes a paragraph on the results/conclusions of the experiments (line 72ff). I think this paragraph would be better placed in the discussion, which seems to lack overall conclusions (it ends rather abruptly).

Over the course of the manuscript, there are many instances where information in my eyes seems to be misplaced. I.e. the remark 'Multiple comparison corrections were made with the cluster method' (line 569), which should be closer to the corresponding statistical test. Additionally, a cluster method does not correct for multiple comparisons, although a permutation test does. While the intent is clear to me, the overall wording is misleading and more details on the procedure should be provided.

Minor comments:

- Line 514ff: 'Three participants used the average of T7 and T8 electrodes for re-referencing as the earlobe electrodes were extremely noisy' – This is an odd choice since in my experienced both of these electrodes are typically very noisy.
- Were the correlation analyses corrected for multiple comparisons as well?
- I would like to see the actual time-frequency representations instead of just the plotted t-values

Version 1:

Reviewer comments:

Reviewer #1

(Remarks to the Author)

This is a revision of a paper I had reviewed earlier. In my previous review I mainly asked for the inclusion of relevant previous research on this topic, and a clarification of some procedural details. I think the authors have taken these points on board appropriately. I would have leaned towards acknowledging previous research in the intro rather than the GD, but this is probably a matter of taste or even format constraints of this journal. To conclude, I think the paper is ready to go.

Reviewer #2

(Remarks to the Author)

I have seen and commented on the previous version of this manuscript. The authors have addressed my previous concerns, and so I consider the paper ready to go.

Signed, Bernhard Hommel.

Reviewer #3

(Remarks to the Author)

I thank the authors for their elaborate and detailed responses (also to the other reviewer's remarks) and highly appreciate the amount of work put into this revision (for example replicating the findings on re-referenced data). The changes to the manuscript very much improved the readability while also adding important details.

I have no further remarks.

Response letter

Page 1-4, response to reviewer 1

Page 5-6, response to reviewer 2

Page 7-10, response to reviewer 3

(Author responses are in bold)

Reviewer #1 (Remarks to the Author):

This paper seeks to find evidence for anticipatory codes of action effects, hence predictable perceptual feedback of motor activities (in this case keypresses). When participants were asked to report the timepoint of an action and its predictable auditory effect on a Libet clock, they perceived the action to occur later than when the same action produced no effect (temporal action binding). Importantly, this action binding went along with a shift of visual attention towards the position of the pointer at which an auditory effect predictably occurred. This attention shift was apparent already 1 s before action onset in detection performance for small visual stimuli around the Libet clock and it was apparent even earlier in EEG data.

This is an interesting manuscript. In fact, it is tricky but would be theoretically important to find precursors of perceptual feedback codes in action production. The paper does so in a clever way by combining and building on various sources of evidence. All methods appear very sophisticated to me. I have a few comments the authors may consider.

1) There have been other attempts to track the ‘unobservable’ (codes of forthcoming perceptual feedback). One approach relied on frequency tagging such effects, and checking whether there are power changes in the EEG corresponding to forthcoming action effects prior to action execution (Dignath et al., 2020 JEP:G). Another approach relied on anticipatory eye movement towards the location of upcoming action effects (Pfeuffer et al., 2016, JEP:G). The present approach is markedly different and novel, but the pros and cons might be discussed somewhere.

We would like to thank the reviewer for suggesting the interesting and highly relevant studies. The papers mentioned by the reviewers have been discussed in the revised manuscript, which reads as (Line 304-324 in the clean version of the revised manuscript):

‘There is increasing experimental evidence in humans, from both functional magnetic resonance imaging²⁹ and neurophysiological^{30,31} studies, supporting the idea of action-effect pre-activation. For example, when a keypress consistently triggered an SSVEP stimulus at 6 Hz, a reduction in the 6 Hz EEG signal was found before the keypress, which was assumed to be the consequence of action-effect prediction^{30,32}. Our study corroborates previous findings by showing that a lateralisation of alpha power in the visual cortex is a neural signature of action-effect prediction in this particular task. The action-effect was a sound in the present study, but the key feature of the sound relevant to the task performance was the clock hand position at the sound onset (i.e. a visual feature). Therefore, a pre-activation of the action-effect was evident in a visual feature related neural signal. Specifically, alpha oscillations in the visual cortex are widely known for the role of supporting the allocation of spatial attention^{19-21,33}. A lateralisation of alpha

*power before the action onset indicates the pre-activation of the spatial information of the action-effect. This finding is consistent with the observation that action-effect pre-activation operates in the neural structure responsible for sensory processing of the action-effect*²⁹. *Furthermore, our finding is also conceptually consistent with a previous study showing that predictable visual action-effects were associated with increased rates of spontaneous saccades towards the location of action-effects before the action-effects were presented*³⁴. *Importantly, we could exclude the possibility that the alpha lateralisation observed here was due to saccades through careful eye-tracking control.'*

2) It was not perfectly clear to me why there was no attention shift apparent when attention was probed only 150 ms after the keypress, which is a very short delay, although there was still action binding present in that condition (Exp. 3). Also, the EEG study suggest an attention shift rightwards up to 700 ms after keypress onset. So would we not predict that the attention shift was apparent even 150 ms after the keypress in Exp. 3 as well? I think this apparent contradiction (at least in my eyes) should be discussed somewhere.

We indeed did not find an attention shift in the instrumental condition compared to the baseline condition at 150 ms or 350 ms after the keypress in the behavioural data (Exp. 3), as pointed out by the reviewer. However, this does not contradict the fact that there was still action binding there. It was expected that there should be action binding in Exp. 3, as the testing paradigm was the classic action binding task and it was same as in Exp. 1 (apart from the time of attention measure). It was the research question whether there would be an attention shift at 150 ms and 350 ms. The results of the experiment showed no attention shift at both time points, thereby suggesting that the attention distribution at 150 ms or 350 ms did not contribute to action binding in the current testing paradigm.

We should clarify here that in the EEG study, the attention shift was found to be significant in two time windows. The first time window was from -1000 ms to -500 ms, and the second time window was from -350 ms to 100 ms. In the revised manuscript, we marked the two significant time windows using more salient horizontal lines than the ones in the previous manuscript to make the information clearer (Fig. 4c in the revised manuscript, which can also be found below). Therefore, the behavioural results do match the EEG results in terms of the timing of the attention shift.

Fig. 4c in the revised manuscript. The time windows showing significant attention differences were marked with black horizontal lines.

3) I missed one piece of information, but it may have escaped my attention. How long did the pointer rotate before participants made a self-paced keypress? I'm asking because the EEG data suggest a fairly long pre-activation interval. This would make sense only if the trial lasted already that long and did not include events from preceding trials.

The onset of the SSVEP stimuli started about 1400 ms before the keypress (depending the exact keypress time from the participants), and the pointer rotation started about 900 ms before the keypress. The relevant alpha power lateralisation peaked around 1300 ms before the keypress, which was slightly after the onset of the SSVEP stimuli. We think the timing of the alpha power lateralisation makes sense as the onset of the SSVEP stimuli might have led the participants into the task mode, thereby activating task-related information processing. The average inter-trial interval among the participants was around 6 seconds, and the participant with the smallest average inter-trial interval was 4.9 seconds. Therefore, we are confident that the pre-activation interval is not contaminated by preceding trials. Relevant timing information has been clarified/updated in the revised manuscript, which reads as (Line 270-272; Line 493-495):

'Note that the onset of the SSVEP stimuli and the clock was about -1400 ms and -900 ms, respectively. The alpha power difference peak at -1300 ms was right after the onset of the SSVEP stimuli.'

'The average inter-trial interval was 6.64 seconds (SD = 1.22, range = [4.92 9.45]) in the baseline condition and 5.96 seconds (SD = 1.09, range = [4.71 10.04]) in the instrumental condition.'

Please note that we also did not stress that the pre-activation started at about 1800 ms before the action onset in the revised manuscript. This is because we used a 1 second long sliding window to calculate the EEG power. Although the data indeed showed a lateralisation of the alpha power difference starting from about -1800 ms, the alpha power at -1800 ms was actually calculated using the data between [-2300 -1300] ms. In the revised manuscript, we described the alpha lateralisation timing using the time point where the alpha power difference was the strongest (i.e. at about -1300 ms). This should make more sense.

4) Wording. In several places the authors talk about 'pre-activation of action-effects' (e.g. in the discussion, L. 283). I think this is not a 100% correct. What the data very convincingly show is that there is a shift of visual attention towards the position where a rotating pointer would arrive when a predicted auditory effect is presented. This indicates that the time point of the effect is somehow pre-activated. But what else of this auditory event is represented or pre-activated, we don't know. This would require to observe traces of features other than the time point of this auditory event (like pitch or intensity).

We would like to thank the reviewer for raising this critical point, which was not made super clear in the previous manuscript. We totally agree that the action-effect here was a sound and that the pre-activation was not directly related to the auditory feature. However, the sound in our study was inherently associated with a spatial feature. That is, the clock hand position at the sound onset. It was this spatial

feature that the pre-activation refers to, which is also why that the neural signature of the pre-activation was represented in the spatial attention related alpha oscillations. Since the spatial feature was still associated with the sound, it is justified to talk about ‘pre-activation of action-effects’, in our opinion. In the revised manuscript, we have made this point clearer. Relevant parts read as (Line 64-67; Line 311 - 324):

‘As introduced earlier, the critical feature of action-effects in the action binding effect is the spatial location of the clock hand associated with action-effect onset. If the action-effect is pre-activated prior to the action execution, we should see a change in the attention paid to the spatial location associated with the action-effect.’

‘The action-effect was a sound in the present study, but the key feature of the sound relevant to the task performance was the clock hand position at the sound onset (i.e. a visual feature). Therefore, a pre-activation of the action-effect was evident in a visual feature related neural signal. Specifically, alpha oscillations in the visual cortex are widely known for the role of supporting the allocation of spatial attention^{19-21,33}. A lateralisation of alpha power before the action onset indicates the pre-activation of the spatial information of the action-effect. This finding is consistent with the observation that action-effect pre-activation operates in the neural structure responsible for sensory processing of the action-effect²⁹. Furthermore, our finding is also conceptually consistent with a previous study showing that predictable visual action-effects were associated with increased rates of spontaneous saccades towards the location of action-effects before the action-effects were presented³⁴. Importantly, we could exclude the possibility that the alpha lateralisation observed here was due to saccades through careful eye-tracking control.’

Minor observations

P1, l1, Should it read ‘predicted action effects shape...’

L 31. Event-fileS

l. 60 A previous suggested ... the sentence reads odd, something is missing.

L 151 detection rate

L 231 field instead of filed.

L 284, Has shown instead of showed

Thank you for carefully reading our manuscript. The above errors have been corrected in the revised manuscript.

Reviewer #2 (Remarks to the Author):

This is a well-written manuscript about a well-motivated study. The presentation is dense and not always easy to follow, but I think the logic and the sequence of the experiments makes sense, and is informative. The studies were carried out expertly, the findings are clear and interesting, and the discussion is reasonably balanced. My only concern relates to the fact that the authors make false claims regarding precedence: the fact that representations of action effects are activated before the action is initiated has been reported before, both for fMRI and EEG methods, see:

1. Kühn, S., Keizer, A., Rombouts, S.A.R.B., & Hommel, B. (2011). The functional and neural mechanism of action preparation: Roles of EBA and FFA in voluntary action control. *Journal of Cognitive Neuroscience*, 23, 214-220.
2. Dignath, D., Kiesel, A., Frings, C., & Pastötter, B. (2020). Electrophysiological evidence for action-effect prediction. *Journal of Experimental Psychology: General*, 149, 1148-1155.

This should be rectified: claims of precedence need to be downplayed and the remaining novel contribution of the present findings need to be discussed more specifically.

Signed, Bernhard Hommel

We would like to thank Professor Hommel for the valuable feedback. In the revised manuscript, the two highly relevant studies have been included in the discussion section. Relevant parts read as (Line 304-324 in the clean version of the revised manuscript):

'There is increasing experimental evidence in humans, from both functional magnetic resonance imaging ²⁹ and neurophysiological ^{30,31} studies, supporting the idea of action-effect pre-activation. For example, when a keypress consistently triggered an SSVEP stimulus at 6 Hz, a reduction in the 6 Hz EEG signal was found before the keypress, which was assumed to be the consequence of action-effect prediction ^{30,32}. Our study corroborates previous findings by showing that a lateralisation of alpha power in the visual cortex is a neural signature of action-effect prediction in this particular task. The action-effect was a sound in the present study, but the key feature of the sound relevant to the task performance was the clock hand position at the sound onset (i.e. a visual feature). Therefore, a pre-activation of the action-effect was evident in a visual feature related neural signal. Specifically, alpha oscillations in the visual cortex are widely known for the role of supporting the allocation of spatial attention ^{19-21,33}. A lateralisation of alpha power before the action onset indicates the pre-activation of the spatial information of the action-effect. This finding is consistent with the observation that action-effect pre-activation operates in the neural structure responsible for sensory processing of the action-effect ²⁹. Furthermore, our finding is also conceptually consistent with a previous study showing that predictable visual action-effects were associated with increased rates of spontaneous saccades towards the location of action-effects before the action-effects were presented ³⁴. Importantly, we could exclude the

possibility that the alpha lateralisation observed here was due to saccades through careful eye-tracking control.'

As you can see from the above new added text in the revised manuscript, we have followed the advice to downplay the precedence claim and to discuss the contribution of the current study more specifically. In addition, we worked on the whole manuscript again to improve the presentation, and a native English Speaker from our group performed a proof-reading of the manuscript. Hopefully, the revised manuscript is easier to follow.

Reviewer #3 (Remarks to the Author):

The manuscript 'Predicted action-effect shapes action representation through pre-activation of alpha oscillations' by Wang and colleagues deals action-effects in an interesting and innovative way. Overall, I really enjoyed reading it, although I have a few remarks. I feel like some parts of the manuscript are quite challenging and hard to understand. This might be due to the extent of reported experiments and the structure required by the journal (methods at the end of the manuscript). Although the authors seem to acknowledge this and have included quite a bit of methodology in the results section, the manuscript is hard to understand and the line of argument hard to follow. A thorough revision of the manuscript in terms of readability would make the manuscript more accessible for a wider readership.

We would like to thank the reviewer for carefully reading our manuscript. During the revision, we worked on the whole manuscript again to improve the presentation. In addition, a native English speaker from our group did a proof-reading of the manuscript. We hope that the revised manuscript has substantially improved in terms of readability.

In terms of the overall structure, it seems quite odd to me that the Introduction includes a paragraph on the results/conclusions of the experiments (line 72ff). It think this paragraph would be better placed in the discussion, which seems to lack overall conclusions (it ends rather abruptly).

Over the course of the manuscript, there are many instances where information in my eyes seems to be misplaced. I.e. the remark 'Multiple comparison corrections were made with the cluster method' (line 569), which should be closer to the corresponding statistical test. Additionally, a cluster method does not correct for multiple comparisons, although a permutation test does. While the intent is clear to me, the overall wording is misleading and more details on the procedure should be provided.

The summary paragraph at the end of the introduction was required by this journal. In the submission requirements, it is explicated stated that in the introduction section 'the final paragraph should be a brief summary of the major results and conclusions.'

An overall conclusion has been added to the end of the discussion section in the revised manuscript, which reads as (Line 361-365 in the clean version of the revised manuscript):

'To conclude, action-effect prediction led to a shift of visuospatial attention towards the action-effect. The attention shift occurred before the action execution and underlay the temporal action binding effect measured with the clock method. An even earlier lateralisation of alpha oscillations before the action execution predicted the attention shift, which served as the neural process supporting the action-effect prediction.'

We also worked on the methods section so that the information about multiple comparison follows directly the corresponding statistical test. In addition, more details about the cluster-based permutation test have been added as suggested by the reviewer. Relevant parts in the revised manuscript read as (Line 589-600):

'Multiple comparisons in the statistical analyses above were corrected using the cluster-based permutation approach⁵². In short, clusters based on adjacency in the

frequency or the time-frequency map were first formed with points showing significant differences between conditions (or significant correlations in the analysis below, with $p < 0.05$ before the multiple comparison correction). For each identified cluster, a cluster-level statistic was calculated as the sum of the test statistics at each point (e.g. the t-value in the case of a t-test). The original data were then randomly permuted between conditions assuming no difference (or between participants assuming no correlation in the analysis below) for 1000 times. For each permutation, the largest cluster-level statistic was kept. The original cluster-level statistics with values bigger than the 97.5 percentile or smaller than the 2.5 percentile of the cluster-level statistics obtained through permutation were considered statistically significant (i.e. a two-sided test).'

Minor comments:

- Line 514ff: 'Three participants used the average of T7 and T8 electrodes for re-referencing as the earlobe electrodes were extremely noisy' – This is an odd choice since in my experienced both of these electrodes are typically very noisy.

Thanks for the reminder. During the EEG data pre-processing, we visually checked the data quality using the 'ft_rejectvisual' function in Fieldtrip. This is also why we could identify the three participants with very noisy data from the earlobe electrodes. For the final selected electrodes, we can confirm that they have data quality good enough for the purpose of re-referencing. Although the data quality judgement was a subjective one from the senior author of paper (with over 10 years' experience of MEEG data analysis), the following objective results support the validity of the judgement:

- 1. As stated in the manuscript, after the data re-referencing, trials were excluded if the amplitude range between -1500 ms and 1000 ms exceeded 200 μ V. On average, 147 trials remained out of a total of 160 trials (Line 575 in the revised manuscript). Given a relatively long time window for the calculation of the amplitude range, such a high proportion of remaining trials indicate a good data quality after the data re-referencing.**
- 2. We performed the same data analysis using data re-referenced to the average of all the 64 on-scalp electrodes. The results replicated all the findings with the earlobe referenced data. That is, significant differences in the alpha band power were found between the instrumental and the baseline condition using the 3 second data before the keypress onset (Figure R3-1a), significant differences were also found in the time-frequency plot with a similar alpha lateralisation in a similar time window (Figure R3-1b), and significant correlations were found between the alpha power difference and the attention difference measured with SSVEP (Figure R3-1c,d).**

Therefore, we choose to keep the original data analysis in the revised manuscript.

Figure R3-1 The results of alpha power analysis using average referenced data. The results shown here are very similar to the results shown in Figure 5 of the revised manuscript, which is based on the data referenced to earlobe electrodes.

- Were the correlation analyses corrected for multiple comparisons as well?

Yes. Relevant information has been added to the revised manuscript, which reads as (Line623-624)

'Multiple comparisons were corrected using the cluster-based permutation approach⁵².'

- I would like to see the actual time-frequency representations instead of just the plotted t-values

For the time-frequency plot in Figure 5b of the revised manuscript, it is very difficult to show the actual time-frequency difference between the two conditions without a baseline correction to the time-frequency data. This is due to the inherent 1/f feature in the EEG data (i.e. the power decreases exponentially with increases in frequency). However, the comparison we are interested in is exactly in the time window before the keypress, thus making a choice of the baseline time window very tricky.

Since the stronger power in the instrumental condition compared to the baseline condition is relative, it is possible that both conditions actually had an alpha power decrease in the relevant time window and that the instrumental condition just decreased less. Therefore, we plotted the temporal dynamics of the alpha power (8-12 Hz) in the revised manuscript (Figure 5c) to address the concern from the reviewer. As the reviewer can see, both conditions had an alpha power increase at around 1500 ms before the keypress. Importantly, the alpha power increase was much stronger in the instrumental condition. This supports our claim in the paper

that an alpha power increase in the right visual field actively inhibits visuospatial attention to the left visual field.